# Mode Characterization and Sensitivity Evaluation of a Surface Acoustic Wave (SAW) Resonator Biosensor: Application to the Glial-Fibrillary-Acidic-Protein (GFAP) Biomarker Detection

**DOI:** 10.3390/mi14081485

**Published:** 2023-07-25

**Authors:** Antonio Matteo Passeri, Francesco Lunardelli, Daniele Cavariani, Marco Cecchini, Matteo Agostini

**Affiliations:** 1INTA S.r.l., Intelligent Acoustics Systems, Via Nino Pisano 14, 56122 Pisa, Italy; a.passeri1@studenti.unipi.it (A.M.P.); francesco.lunardelli@sns.it (F.L.); d.cavariani@intasystems.net (D.C.); marco.cecchini@nano.cnr.it (M.C.); 2Dipartimento di Fisica, Università di Pisa, Largo B. Pontecorvo 3, 56127 Pisa, Italy; 3NEST, Istituto Nanoscienze-CNR and Scuola Normale Superiore, Piazza San Silvestro, 56127 Pisa, Italy

**Keywords:** biosensors, Rayleigh surface acoustic wave (SAW), shear-horizontal SAW, lithium niobate, finite element modeling, GFAP biomarker

## Abstract

Biosensors based on surface acoustic waves (SAWs) offer unique advantages due to their high sensitivity, real-time response capability, and label-free detection. The typical SAW modes are the Rayleigh mode and the shear-horizontal mode. Both present pros and cons for biosensing applications and generally need different substrates and device geometries to be efficiently generated. This study investigates and characterizes SAW resonator biosensors on lithium niobate in terms of modes generated and biosensing performance. It reveals the simultaneous presence of two typical SAW modes, the first around 1.6 GHz and the second around 1.9 GHz, differently polarized and clearly separated in frequency, which we refer to as slow and fast modes. The two modes are studied by numerical simulations and biosensing experiments with the glial-fibrillary-acidic-protein (GFAP) biomarker. The slow mode is generally more sensitive to changes in surface properties, such as temperature and mass changes, by a factor of about 1.4 with respect to the fast mode.

## 1. Introduction

Biosensor research has attracted considerable attention in the last decades, especially in drug discovery, biomedicine, defense, and security [1]. A biosensor is a device that uses specific biochemical reactions to produce a measurable signal correlated to the concentration of the target analyte, such as glucose, nucleic acids, proteins, virus, and bacteria. Usually, this analyte is contained in a liquid solution. The biosensor incorporates a biological sensing element that generates a measurable signal from biological interactions. A typical biological interaction can be the binding event between the antibody and the target antigen [2]. Antibodies are deposited onto the surface of biosensors through a process called functionalization. This process enables the device to react to interactions between the antibodies and the target molecules in the sample, leading to a specific detection response. The biosensor can provide qualitative or quantitative data depending on the correlation mechanism between the response and the amount of bio-analyte detected. The main scope of biosensors is to provide rapid, accurate, and reliable information about the analyte concentration contained in the sample, ideally in real-time. From the technological point of view, the essential characteristics of biosensors are stability, sensitivity, selectivity, reproducibility, cost, and size. These parameters compete with each other and are generally chosen after a cost-benefit analysis depending on the specific application.

Depending on the transduction mechanism, biosensors can be divided into optical, electrochemical, and mechanical. Optical sensors have advantages, such as high sensitivity, and disadvantages, as they suffer from bulky and expensive readout instrumentation [3]. In contrast, electrochemical sensors are generally less sensitive than optical sensors but smaller and more cost-effective [4]. Mechanical sensors usually offer performance and cost in between the other groups [5]. Acoustic wave biosensors are a subset of mechanical biosensors that exploit acoustic waves as the transduction mechanism. Acoustic wave biosensors can be divided into two main groups according to the type of acoustic mode: bulk acoustic wave devices (BAW) and surface acoustic wave devices (SAW) [6]. In BAW devices, the acoustic wave spreads through the whole volume of the substrate. In SAW devices, the acoustic wave travels on the surface of the substrate, with or without guiding layers. SAW devices can work in the ultra-high-frequency (UHF) range, commonly identified in the 300 MHz–3 GHz range. Higher frequencies lead to more sensitive devices due to the reduced penetration depth of the acoustic wave into the bulk [7]. Thus, UHF-SAW devices are very sensitive to substrate surface modifications, such as mass loading, changes in conductivity, and viscosity [8], eventually overcoming the sensitivity limitation of mechanical sensors while keeping the low cost.

A possible detection technique that can be used for detecting surface changes in SAW biosensors is the resonant frequency shift technique. This approach involves measuring the variation in the resonance frequency of SAW devices. The interactions on the surface lead to a change in the surface density and, thus, to a change in the SAW propagation velocity. This variation can be associated with the concentration of molecules in the sample, thus giving a quantitative analysis. The first attempts to use SAW for biosensing were unsuccessful, as these devices could not function effectively when immersed in fluids. This is because the typical SAW mode, called Rayleigh mode, has a strong displacement component normal to the surface of the substrate. This normal component scatters energy into the fluid and causes pressure waves to be radiated into the liquid. As a result, the SAW is greatly damped and can no longer be exploited for efficient sensing [9]. To overcome the high attenuation issue, the SAW can be generated as polarized horizontally with respect to the substrate surface. These waves are called shear-horizontal SAW (SH-SAW). SH-SAW devices can be used for measurements also in liquids due to the low coupling of SH-SAW with the solution to be analyzed [10]. However, they also have disadvantages, such as a low signal-to-noise ratio and limits in detection performance due to their diffusion into the bulk [11]. Rayleigh SAW devices, on the other hand, have a higher sensitivity to surface modifications. Recently, SAW technology has been successfully exploited for biosensing even in a liquid environment after a drying step [12]. Moreover, Rayleigh SAW technology can efficiently perform biosensing and fluid manipulation tasks [13]. Fluid agitation, mixing, droplet displacement, atomization, and particle displacement within a droplet are all possible with this technology, down to the nanoliter scale [8,14,15]. Due to the advantages of SAW devices, such as low cost, small size, ease of assembly, and the ability to work in the UHF range, they have the potential to transform the field of biosensing, where the detection kinetics can be disregarded [10].

Several pathologies would benefit from such biosensing technologies. Traumatic brain injuries (TBIs) are one relevant example. Over time, the incidence of traumatic brain injury is increasing significantly [16]. Early diagnosis is essential to improve the patient’s clinical condition. Techniques used for quantification, such as computed tomography (CT), are time-consuming, expensive, and increase the risk of radiation exposure [17]. Developing a new portable device that can aid the diagnosis of TBIs is an attractive alternative. To this purpose, glial-fibrillary-acidic-protein (GFAP) has recently become one of the most popular circulating biomarkers for diagnosing TBIs. GFAP levels can reflect the clinical severity and extent of intracranial pathology after TBI [18]. Therefore, GFAP may be important for diagnosing TBI and other pathologies, intracerebral hemorrhage, or neoplastic diseases, such as glioblastoma multiforme (GBM), are a few examples.

In this work, we characterize UHF-SAW resonator biosensors in terms of acoustic modes generated and biosensing capabilities. We exploit finite element modeling (FEM) techniques to predict the mode behavior for sensing mass adhesion. We then characterize the modes’ sensitivity to changes in surface properties, such as temperature and mass changes, and finally test our devices to detect the GFAP biomarker.

## 2. Materials and Methods

### 2.1. Ultra-High-Frequency Surface-Acoustic-Wave Resonator Biosensors

The substrate of our chips was 128° Y-cut X-rotated lithium niobate (LN), which was used because of its high piezoelectric coupling constant and allowed efficient SAW generation. A Ti/Au 10/100 nm layer was deposited on the LN surface and then patterned with standard lithographic techniques and lift-off processes to realize the UHF-SAW resonators. The resonators’ geometry was optimized in previous works [11,13,19] according to the fabrication process capability and desired performance. Every chip comprised six identical resonators: A, B, C, D, E, and F. The resonators were made of a comb-like metal structure called interdigital transducer (IDT) for the wave generation and two grating reflectors with the same periodicity for the wave confinement and enhancement over the IDT area. An oscillating radio frequency (RF) signal is applied to one set of electrodes of the IDT while the other is kept at ground voltage. The six 1-port UHF-SAW resonators served as biosensors. The finger width was λ/4 = 0.4 μm (i.e., λ = 1.6 μm), 50% metallization ratio, with λ being the SAW wavelength. All the devices were thoroughly dried before measurements to avoid wave attenuation into the liquid.

### 2.2. Functionalization and Detection Protocols

SAW sensors generally cannot detect analytes specifically if not appropriately functionalized with probe molecules. Indeed, the sensors’ surface was functionalized with monoclonal anti-GFAP antibodies as probe molecules. In addition, polyclonal anti-GFAP antibodies were used to increase the mass of each bound GFAP further. Every reagent was purchased from Sigma-Aldrich if not otherwise stated. The functionalization of the chips followed this protocol:cleaning of chips performed by sonication in acetone (ACE), isopropanol (IPA), and deionized (DI-) water for 7 min each;drying of the chips with a stream of nitrogen;half-antibody functionalization incubated for 30 min with a solution made of 2 μL pAb (anti-GFAP polyclonal antibody, Synaptic System, 173-002), 4 μL DTT (DL-Dithothreitol 5 g, by SIGMA ALDRICH) at a concentration of 1.5 mg/mL, 34 μL PBS 1X (phosphate buffer saline);wash in DI-water for 5 min;drying the chips with a stream of nitrogen and waiting 1 h before use.

The GFAP detection or PBS negative control with signal amplification followed this protocol:recombinant GFAP (Synaptic Systems, 173-0P) 1000 nM in PBS, or clean PBS for the negative control, incubated for 30 min;wash in DI-water for 5 min;drying of the chips with a stream of nitrogen;pAb for 30 min;wash in DI-water for 5 min;drying of the chips with a stream of nitrogen.

### 2.3. Measurement Setup and Acquisition Protocol

An RF probe station was used to acquire signals (Everbeing Int’l Corp., Hsinchu, Taiwan, model C-6). The probe station was equipped with a temperature-controlled chuck where the chips were placed to thermalize and then take measurements with probe tips. Every measure was taken at 30 °C, if not otherwise stated, after 20 min of thermalization. The S11 spectrum of the UHF-SAW resonators was measured using a vectorial network analyzer (Agilent E5071C). Each spectrum had a 100 MHz span with 6.7 kHz resolution (15,001 points) and was centered at 1590 MHz for the slow mode and 1893 MHz for the fast mode. The collected data were analyzed with custom-made data analysis software that used a cross-correlation algorithm to calculate the resonance frequency shift. Then the average values of the different resonance frequency shifts and the respective standard deviation and standard error of the various measurements were calculated with OriginLab software.

## 3. Results and Discussion

### 3.1. Finite-Element-Modelling Simulations

We used COMSOL Multiphysics to perform simulations to study the properties of the acoustic modes. We developed a three-dimensional (3D) model of a unit cell of the UHF-SAW resonator. The device was simplified by exploiting its periodicity characteristics to reduce the computing power required. As shown in Figure 1a, the 3D model consisted of the unit cell of the IDT and the LN substrate. The unit cell dimensions were length and width λ, height 10 λ. The materials’ elastic stiffness matrix, piezoelectric coupling matrix, and relative electrical permittivity were modified using the built-in Euler rotation approach to match the rotation of the substrate’s cut. A single pair of fingers was designed on the surface to simulate an entire device and a periodic boundary condition was assigned. The periodic repetition of a single pair of fingers was achieved by coupling opposite sides of the unit cell. The spatial discretization was accomplished by creating a mesh over the entire surface, Figure 1a, with a maximum distance from the nodes of 320 nm. One finger was electrically grounded while we applied an RF power of 1 W to the other. We conducted a frequency analysis of the model to verify the coupling with acoustic waves. Figure 1b shows the RF power reflection spectrum (S11). Dips in the S11 were hints of the generation of acoustic waves. The simulations showed two major dips in the S11, at 1666 MHz and 1870 MHz. These results were also confirmed with the COMSOL eigenmodes solver, Figure 1c,d. The 1870 MHz dip was the closest to the resonance frequency of the SAW, even if the resonance frequency found was significantly lower than that predicted by the f_0_ = v_0_/2*p* approximation, where v_0_ is the propagation speed of the SAW (~3980 m/s for LN), while *p* is the periodicity between the electrodes which is equal to λ/2. This could be due to the decreased SAW speed at higher frequencies [20]. Indeed, it is reported in the literature that the SAW speed on LN can reach values also below 3000 m/s at high frequency [21]. In addition, using gold fingers in the UHF-SAW resonators, which are heavier than the usual aluminum ones, could explain the reduced SAW speed. On the other hand, the 1666 MHz dip was unexpected and represented an unexplored acoustic mode. For simplicity, we will refer to the 1666 MHz mode as slow and the 1870 MHz mode as fast.

To understand the spatial configuration of the two modes, we analyzed the displacement distribution over the crystal’s X, Y, and Z directions, as depicted in Figure 2. The X component was the direction of propagation of the SAW, therefore the longitudinal component, while Y and Z were the transversal components, shear and normal, respectively. Figure 2a,b show the component percentage of the total displacement in the resonance frequency interval. The total displacement was 2–20 nm, compatible with typical SAW amplitudes [19]. We observed that both the slow and fast modes showed polarizations almost constant over the resonance interval, except for a narrow frequency region with a strong energy exchange between the components. We called this region the exchange interval (EI), highlighted in blue in Figure 2. Far from the EI, both slow and fast modes were mainly polarized along the transversal components Y and Z, respectively. The slow mode had a stronger shear component Y. In comparison, the fast mode had a stronger normal component Z. This suggested that, far from the EI, the fast mode was mainly a Rayleigh mode, as expected, while the slow mode was mainly shear-horizontally polarized. The simultaneous presence of shear and Rayleigh modes on the LN substrate is little studied. To the best of our knowledge, the present work is the first to report the simultaneous presence of these two modes on LN separated in frequency. Intriguingly, inside the EI, there was a strong modification of the polarization of both the slow and fast modes. For the slow mode, the Z component overcame the Y component, while for the fast mode, the opposite occurred. The longitudinal component X was the least affected by the energy exchange. In other words, inside the EI, the slow mode polarization was more similar to a Rayleigh mode, while the fast mode polarization was more similar to an SH-SAW. Therefore, the modes exchanged their behavior with respect to what happened outside the EI.

Since the SAW polarization is expected to significantly affect the sensing performance to surface modifications, we performed several simulations to study the different sensing behavior of the modes upon surface mass adhesion. The surface mass was added isotopically to the mass of the two fingers in the simulations, varying its value from 10 ng/cm^2^ to 10,000 ng/cm^2^. As shown in Figure 2c, we observed that as surface mass adhesion increased, both dips had a redshift, as expected. We obtained the two sensitivities for the added mass, which were −6.12 kHz/(ng/cm^2^) for the slow mode, and −3.69 kHz/(ng/cm^2^) for the fast mode, respectively. Intriguingly, the slow mode was more sensitive to mass adhesion with a 1.66-fold enhancement with respect to the fast mode.

### 3.2. UHF-SAW Biosensors

The two modes were observed experimentally with the RF measurement setup (Figure 3). The slow mode was found on average at (1577.44 ± 0.02) MHz, and the fast mode was found on average at (1890.28 ± 0.01) MHz, in good agreement with the FEM results. We performed a temperature characterization of the devices to understand the modes’ behavior in response to changes in the propagation velocity due to heating. The temperature of the probe’s chuck was varied from 30 °C to 40 °C with 2 °C steps. The results of the modes’ sensitivity to temperature are shown in Figure 3d. We calculated the mean values of the shift of the resonance frequencies with their respective standard deviation. Then we performed a linear fit of this shift at different temperatures for each sensor. We obtained different temperature sensitivities for the slow and fast modes, (−201 ± 4) kHz/°C and (−156 ± 11) kHz/°C, respectively. The temperature sensitivity for the slow mode was slightly higher than the fast mode, with a 1.3-fold enhancement. Since the effect of temperature variation is to a first approximation similar to that of mass adhesion, as both cause a slowing of the propagation speed of the modes, these results were consistent with those obtained from FEM simulations on mass adhesion. The slow mode was confirmed as the most sensitive wave polarization. No significant difference was observed in the S11 of the sensors at different temperatures, except for the resonance frequency position. The spectrum shape was preserved, suggesting that the energy exchange between the slow and fast modes is not affected by temperature changes in the tested range.

We then investigated how the devices responded to the biosensing experiments. For these experiments, performed on different days, we used four chips (24 UHF-SAW resonator biosensors). We first functionalized the chips using the procedure described in the Materials and Methods section. The results in Figure 4 show the mean shift of the six sensors of each chip and the standard error of the different chips. Measurements were acquired to observe the frequency shifts of the slow and fast modes upon mass adhesion due to the functionalization, as shown in Figure 4a,b. A higher sensitivity of the slow mode than the fast mode was observed. The slow mode average shift to functionalization was −1882 kHz, while the fast mode shift was −1108 kHz. The slow mode shift was 1.70 times larger than the fast mode shift. After functionalizing the chips, we performed biosensing experiments with the GFAP-containing solution. PBS was used as a negative control. Both were then amplified with pAb. As shown in Figure 4c,d, we observed an average resonant frequency shift to the PBS negative control of −180 kHz for the fast mode and −140 kHz for the slow mode. With the GFAP solution, an average shift of −1020 kHz was observed for the slow mode, and −720 kHz was observed for the fast mode. Using the PBS level as a reference, the slow mode shift was, on average, −585 kHz and the fast mode shift was −423 kHz. Although the standard errors were higher in the PBS and GFAP biosensing experiments with respect to the functionalization ones, the average shifts reflected what we observed previously, namely, a higher sensitivity of the slow mode with respect to the fast mode with a 1.38-fold enhancement. It can be inferred that, throughout our numerical analysis and experimental procedures, there was a higher sensitivity to general surface modifications for the slow mode than for the fast mode, with an average factor of about 1.4 between the two.

This behavior, namely the higher sensitivity of the slow mode with respect to the fast mode, could be explained by the results of the FEM analysis in terms of polarizations inside the EI. While outside the EI, the slow mode is more similar to an SH-SAW, and the fast mode is more similar to a Rayleigh SAW. The opposite occurred inside the EI. Since it is known by literature that the Rayleigh SAW mode is more sensitive to surface modifications with respect to the SH-SAW mode, and the slow mode resembles a Rayleigh mode inside the EI, the slow mode was expected to have a higher shift to surface modifications.

## 4. Conclusions

In this work, we investigated and characterized UHF-SAW resonator biosensors in terms of acoustic modes generated and biosensing capabilities. We used FEM techniques to study the acoustic modes of the resonators and predict their behavior for the detection of bio-analytes. We then characterized the sensitivity of the modes to changes in surface properties, such as temperature and mass changes, and finally tested our devices for GFAP biomarker detection. We observed in such UHF-SAW resonators the presence of two modes, a fast mode and a slow mode, and a strong energy exchange among their components inside the EI. We studied these two modes and showed, using FEM analysis and experimentally with temperature changes and the GFAP biomarker, that the slow mode is generally more sensitive than the fast mode to changes in surface properties, with an average increase of about 1.4 times in sensitivity. This could be due to the fact that inside the EI, the slow mode is more similar to a Rayleigh SAW, while the fast mode is more similar to an SH-SAW. Therefore, since Rayleigh waves are more sensitive, a higher signal to surface change was observed numerically and experimentally. To the best of our knowledge, our work is the first to report the simultaneous presence of these two modes on LN at high frequency and to assess their sensing behavior. The presence of the two modes polarizations may facilitate the future design and fabrication of devices that exploit SAW for microfluidics and biosensing on the same chip. Further studies are needed to explain in more detail the sensing mechanisms of these waves in LN or other materials.

## Figures and Tables

**Figure 1 micromachines-14-01485-f001:**
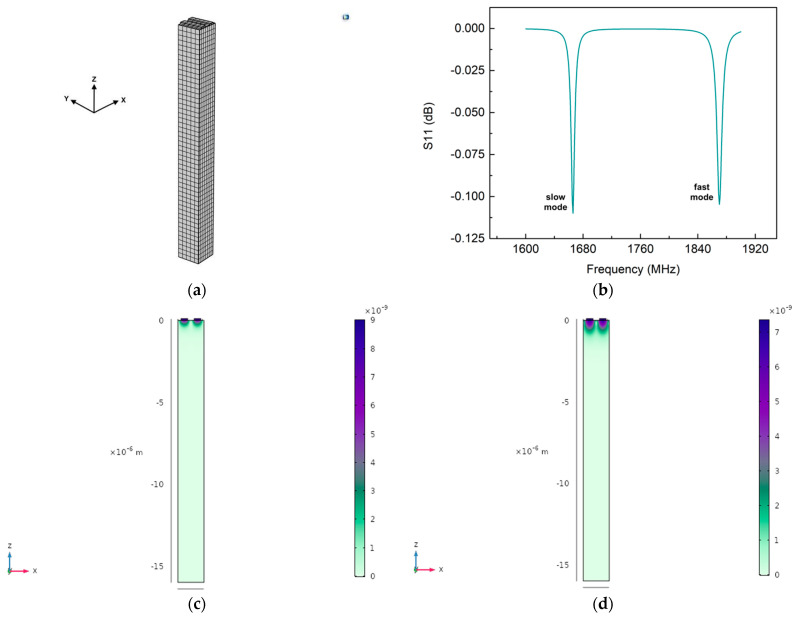
UHF-SAW resonator FEM analysis. (**a**) The 3D model mesh; (**b**) S11 graph of slow mode and fast mode; eigenmodes solutions for the slow (**c**) and fast (**d**) acoustic waves, at 1666 MHz and 1870 MHz, respectively.

**Figure 2 micromachines-14-01485-f002:**
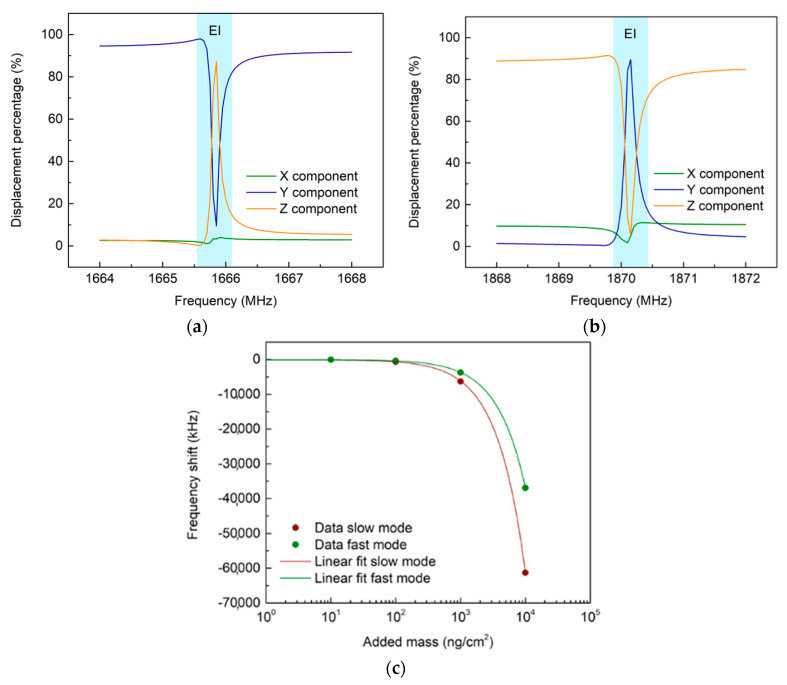
Mode polarization and mass sensing simulation. (**a**,**b**) Polarization percentage of slow and fast modes, respectively (the EI is the blue-shaded area); (**c**) mass-adhesion simulations with the linear fit (semi-logarithmic graph).

**Figure 3 micromachines-14-01485-f003:**
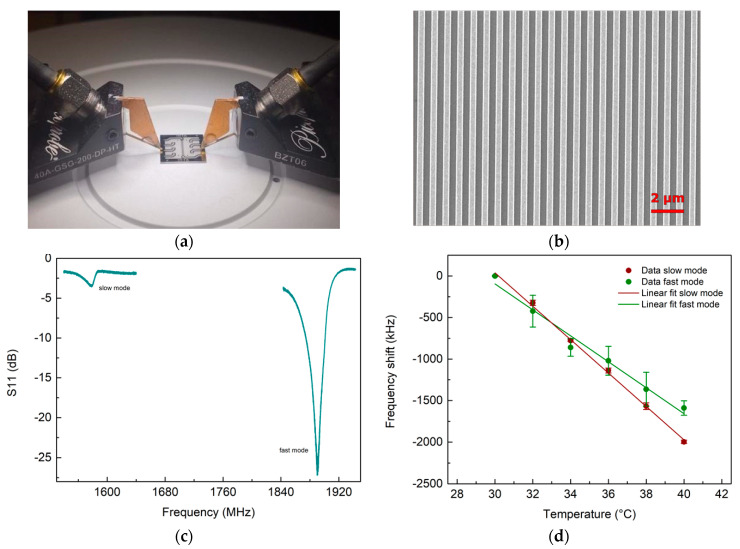
RF setup and temperature characterization of the UHF-SAW resonators. (**a**) UHF-SAW chip under test; (**b**) scanning-electron-micrograph of a resonator; (**c**) typical S11 measured by the RF setup; (**d**) temperature sensitivity for the slow and fast modes.

**Figure 4 micromachines-14-01485-f004:**
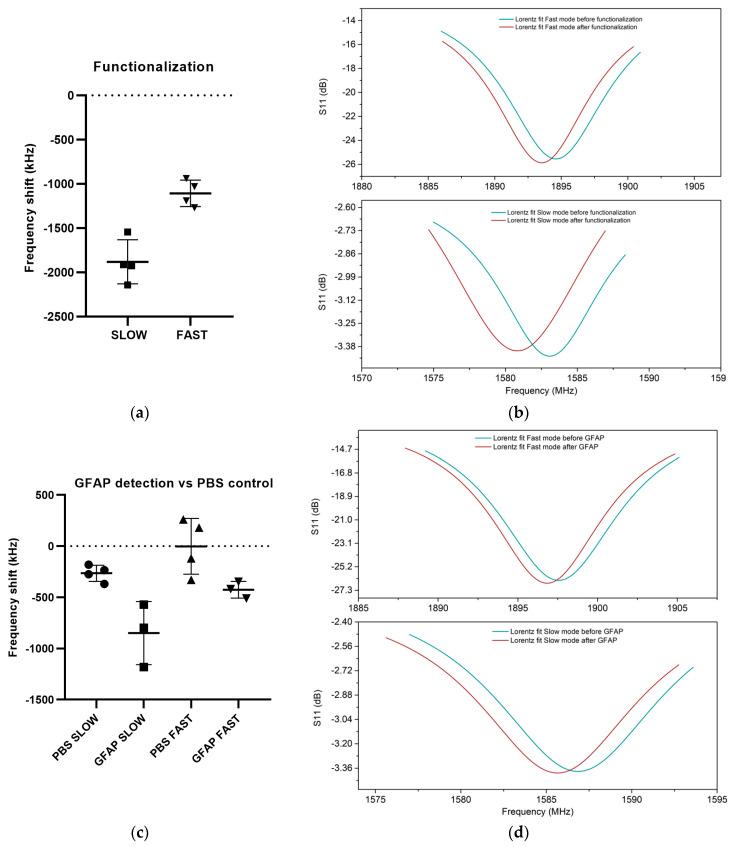
Biosensing experiments with the UHF-SAW resonators. (**a**) Frequency shift to the functionalization for the slow and fast modes. The averages and standard errors of the various chips are shown. (**b**) Slow mode and fast mode shift. A Lorentz fit of sensor A was performed before (green) and after (red) the addition of the functionalization. (**c**) The frequency shift of the GFAP and the PBS control for the slow and fast modes. The averages and standard errors of the various chips are shown in the figure. (**d**) The slow mode and fast mode shift. A Lorentz fit of sensor A was performed before (green) and after (red) the addition of GFAP and polyclonal antibody.

## Data Availability

Data available at https://zenodo.org/record/7937265.

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
