# Peer review of "Mode Characterization and Sensitivity Evaluation of a Surface Acoustic Wave (SAW) Resonator Biosensor: Application to the Glial-Fibrillary-Acidic-Protein (GFAP) Biomarker Detection"

_micromachines, 2023, doi:10.3390/mi14081485_

Round 1

Reviewer 1 Report

The research mentioned in this article has certain significance in terms of acting and is novel and interesting. It is recommended that the author include more research data to enhance persuasiveness. Specific suggestions are as follows:

1. The abstract section is not concise and direct enough. Please reorganize the language and make modifications.

2.  There are too many keywords, and they need to be reduced in number.

3. There are numerous issues with the reference list. Please revise each item accordingly.

4. The layout of the author's images is chaotic and needs to be rearranged. Please do not place the numbering below the images.

5. The author's experiments are not comprehensive. The author did not conduct a series of characterizations and parameter optimizations for the electrodes. Additionally, the author needs to further investigate the performance of the sensor, such as its interference resistance capability.

Author Response

We thank the Reviewer for the comments. We believe the manuscript will be improved by following the Reviewer's suggestions. Below is a point-by-point reply to the Reviewer's comments. The modifications to the text are highlighted in yellow in the revised version of the manuscript.

  1. The abstract has been shortened and made more schematic and direct.
  2. The number of keywords has been reduced.
  3. The reference list has been reviewed.
  4. We improved the figures' clarity and removed the numbers below them.
  5. We thank the Reviewer for this pertinent comment. The electrode optimization of our device in terms of fabrication and sensing performance has been carried out in our previous works, such as ref. [DOI: 10.1002/adfm.202201958; DOI: 10.1109/ACCESS.2019.2919000; https://doi.org/10.1016/j.snb.2017.07.014] cited in the revised manuscript. Further improvements are beyond the scope of the present work. Nonetheless, we better clarified this point in the main text of the revised manuscript.

Reviewer 2 Report

While the paper presents an interesting finding about using SAW resonators as biosensors to detect GFAP, it is not publishable in its current form due to several significant issues.

1.       It is well-known that SAW devices can be used as sensors because the mass loaded onto the sensor alters the SAW properties and resonant frequency. However, SAW devices do not have the ability to selectively choose analytes. Therefore, what people usually do is coating a layer of specialized material as a sensing layer to selectively target the desired analyte. In this regard, the authors' claim of detecting GFAP lacks credibility, as their devices are incapable of selectively choosing GFAP; they simply detect whatever is attached to the devices.

2.       The simulation is completely wrong. In the authors’ case, people don’t simulate S11. What the authors want and need to do first is actually to use the eigenmode solver in COMSOL. If the claimed slow or fast modes indeed exist, they can be found within the eigenmode solutions. Please remember to include the mode profiles in the paper. The current model and solver being used by the authors cannot help in finding any mode. For example, without the energy dissipation part, waves reflect everywhere and make the solution a mess. This might be one of the reasons why the simulated S11 result (figure 1b) appears odd - the lowest S11 is only about -0.1db and the frequencies are very different from the experiment results. These discrepancies are likely the result of random coincidences within the simulation.

3.       The experiment results obtained are very odd. The wavelength of the wave is 1.6um, which is determined by the IDT. Multiplying it by the frequencies of the two modes, ~1600 MHz and ~1800 MHz, it can be deduced that the wave speeds are approximately 2.56km/s and 2.88km/s. However, on a 128° Y-cut X-rotated lithium niobate (LN) substrate, the slowest wave mode, the Rayleigh wave, typically exhibits a speed of approximately 4 km/s. Additionally, exciting the SH-wave mode on this particular LN sample is very challenging. The discrepancies raise doubts about the validity of the experimental results.

4.       It is essential to include images (such as SEM) of the SAW resonator devices to provide visual details and enhance the paper's comprehensibility. Additionally, improvements are needed in the S11 measurements.

While there may be other minor issues present in the paper, they are not significant enough to warrant mention at this stage. Addressing the major concerns is crucial before considering the paper suitable for publication.

Author Response

We thank the Reviewer for the comments. We believe the manuscript will be improved by following the Reviewer's suggestions. Below is a point-by-point reply to the Reviewer's comments. The modifications to the text are highlighted in yellow in the revised version of the manuscript.

  1. The Reviewer is correct in saying that SAW sensors, like the ones presented in our work, are mass sensors. Therefore, SAW sensors cannot specifically detect analytes if not appropriately functionalized with probe molecules. Indeed, in our work, we functionalize the sensors' surface with monoclonal anti-GFAP antibodies as probe molecules. Moreover, we use polyclonal anti-GFAP antibodies to increase the mass of each bound GFAP further. Thus, the selectivity to GFAP is provided by both antibodies, as also shown in Figure 4, where the GFAP detection and negative controls are present. Nonetheless, we better clarified this point in the Materials and Method section.
  2. We thank the Reviewer for this comment. We believe that confirming our results with different approaches is an added value to the manuscript's consistency. In fact, we performed an eigenvalues study in COMSOL, as suggested. The results match our previous S11 analysis, which, indeed, was correct. We added this information to the main text, along with the modes displacement in Figure 1 of the revised manuscript.
  3. The Reviewer is correct in saying that the SAW speed we observed was lower than expected, as we also point out in the main text. The Rayleigh-SAW speed (fast mode) obtained by our measurements was approximately 3000 m/s, while it is usually around 4000 m/s. This low SAW speed may be due to two facts. First, the SAW speed significantly decreases in LN at high frequency, as also reported in [J. Vac. Sci. Technol. B 33, 06F401 (2015); https://doi.org/10.1116/1.4935561] which show results compatible with our findings, where the SAW speed reaches values also below 3000 m/s (cited in the revised manuscript). Second, we used gold fingers in our resonators, which are known to decrease the SAW speed more than the usual aluminum, since the gold density is almost 10x higher than the aluminum one. We better clarified this point in the main text.
  4. We agree with the Reviewer that SEM images will improve the manuscript's understandability. Therefore, we included them in Figure 3.

Reviewer 3 Report

The manuscript presents some interesting consideration on the two oscillation modes within a SAW sensors. They are presented both from theoretical and experimental perspectives. The study is interesting for the readers, in my opinion, but I suggest two minor changes to the manuscript:

1) Introduction:

“The scientific field of biosensing is based on several disciplines. Depending on the transduction mechanism, biosensors can be divided into optical, electrochemical, and mechanical. The interaction of the optical field with a biological element is exploited for the operation of optical sensors. They have advantages, such as their high sensitivity and disadvantages, as they suffer from bulky and expensive readout instrumentation [3]. In contrast, electrochemical sensors take advantage of the enzymatic catalysis of a reaction involving electron exchange. They are generally less sensitive than optical sensors but smaller and more cost-effective [4]. Mechanical sensors usually offer performance and cost in between the other groups [5]. Acoustic wave biosensors are a subset of mechanical biosensors that exploit acoustic waves as the transduction mechanism.”

  • I wonder if the paragraph presence is appropriate. Is it a ‘review article’ or a ‘scientific paper’ after all ?

2) Fig. 3d); Is any relation between the intersection points of the two curves and the ‘switch’ in between the ‘low’ and ‘high’ waves as Rayleigh and shear waves ? Could authors comment on the ‘switched behavior’ and the “thermal” behavior of the two characteristics ? Is ‘switching’ happening regardless of the device working temperature ? I think that this discussion would be interesting for the readers.

Author Response

We thank the Reviewer for the comments. We believe the manuscript will be improved by following the Reviewer's suggestions. Below is a point-by-point reply to the Reviewer's comments. The modifications to the text are highlighted in yellow in the revised version of the manuscript.

  1. We agree with the Reviewer and shortened the paragraph.

  2. We agree with the Reviewer and added a more comprehensive discussion in Figure 3d. The crossing point between the linear fits of Figure 3d may not have any physical insight. We believe it is merely due to the fit intercept that does not align perfectly with the zero point on the plane.

Reviewer 4 Report

This article conducted simulation and experimental research on a 128LN based surface acoustic wave biosensor, but from the content of the paper, there are problems such as insufficient innovation and unclear concepts.

First of all, the so-called ultra high frequency SAW is not well described. The frequency in this paper is about 1.5GHz, which belongs to the general category in the field of SAW;

Secondly, the paper uses 128LN as the piezoelectric substrate and gold as the electrode, which does not result in slow or fast waves. In fact, typical Rayleigh waves and shear waves coexist. Rayleigh wave velocity is slower than shear wave velocity. The paper needs to be presented in a more accurate way;

In addition, the paper uses gold electrodes and 128LN devices for biosensing, and the Rayleigh wave mode should be difficult to observe the signal, as the liquid environment will cause severe longitudinal coupling attenuation in Rayleigh wave; Shear wave mode may be used for biosensing, but surface electrodes generally use protective layers such as SiO2, otherwise damage may occur during the sensing process.

Finally, the use of shear type surface acoustic waves excited by a gold electrode on 128 LN for biosensing does not seem to have significant advantages compared to the widely used Love wave mode.

Author Response

We thank the Reviewer for the comments. We believe the manuscript will be improved by following the Reviewer's suggestions. Below is a point-by-point reply to the Reviewer's comments. The modifications to the text are highlighted in yellow in the revised version of the manuscript.

  1. The frequency range of 300MHz-3GHz is widely recognized as the ultra-high-frequency (UHF) range. As our devices' resonance frequencies fall within this range, we consistently refer to them as UHF sensors, as we have done in our previous works. To avoid any ambiguity, we will provide a clearer explanation of this point in the introduction section of the manuscript.
  2. In previous literature, the co-existence of shear and Rayleigh waves on LN128 has not been adequately described. Despite our efforts, we could not locate any evidence of distinctly separate Rayleigh and shear modes in LN128 from other studies. However, we are eager to include any relevant references if the Reviewer can provide us with suggestions or references that support the existence of such modes.
  3. The Reviewer correctly points out that the Rayleigh wave would be attenuated in a liquid medium. We want to emphasize that all our measurements were conducted after a thorough drying step of the resonators, as stated in the experimental protocols. Nonetheless, we will provide a more explicit clarification of this point in the Materials and Methods section to avoid any confusion.
  4. We provided evidence of the simultaneous presence of shear and Rayleigh waves on the same substrate, propagating in the same direction but clearly separated in frequency for the first time to the best of our knowledge. This discovery holds significant potential for various fields, including SAW microfluidics and biosensing, where different wave polarizations are typically employed on various substrates. Our findings have the potential to inspire new studies, enabling easier device design and fabrication. To ensure the significance of this point is well-communicated, we will provide a more explicit clarification in the conclusions section of the manuscript.

Reviewer 5 Report

The authors present a manuscript on biosensing performance and comparison of high-order modes in SAW devices.  The mansucript is well written and reports a useful finding to the field of mechanical biosensors for which there is considerable interest in the use of high-order modes for detection applications.  I recommend the manuscript may be published pending these minor revisions:

1) I suggest to edit the manuscript to better clarify that the terms "slow and fast" modes are nomenclature created by the authors.  For example in the abstract, i suggest the author revised the text to say, ... "differently polarized and separated in frequency, 'which we refer to as' slow and fast modes."  

2) The authors compare the performance high-order modes based on their nomenclature "slow vs. fast".  However, this is somewhat confusing to readers interested in the mode shape and the frequency of the mode.  I suggest the authors may also include the frequency of the modes and mode types/shapes in the abstract when comparing performance. The authors may simply want to adopt the commonly used nomenclature of "low" and "high" order modes. 

Author Response

We thank the Reviewer for the comments. We believe the manuscript will be improved by following the Reviewer's suggestions. Below is a point-by-point reply to the Reviewer's comments. The modifications to the text are highlighted in yellow in the revised version of the manuscript.

  1. We agree with the Reviewer and modified the text accordingly.
  2. We thank the Reviewer for this pertinent comment. We would prefer to keep the "slow" and "fast" names of the modes because they directly recall the wave speed, which is a key aspect of the physics underlying biosensing. Nonetheless, we better clarified the nomenclature in the abstract as suggested to avoid any confusion.

Round 2

Reviewer 1 Report

no comments

Author Response

We thank the Reviewer for the valuable comments received on our work.

Reviewer 2 Report

The manuscript cannot be approved for publication due to the same reasons I mentioned previously.

1.       The simulations are wrong. Primarily, the simulation of S11 is not feasible through COMSOL. S11 is one of the scattering parameters. It reflects the power that is bounced back from the source or input port, and it is directly related to the device's impedance. To illustrate, if the device has an impedance of 50 Ohm (a standard in microwave engineering), which is the same as your VNA, the S11 value would be 0, indicating no power reflection and complete power transmission. This property of the device is fully determined by its resistance and frequency-dependent capacitance and inductance. These cannot be derived from a simplified COMSOL model. In addition, there is no such thing as S11 spatial distribution. Additionally, if the authors employed an eigenmode solution, they would only identify a limited number of eigenmodes with discrete frequencies. The frequency sweep is impossible. Given the constraints of a crystal's stress/strain tensors due to its symmetry, the number of eigenmodes is significantly limited and these eigenmodes have specific names, such as Rayleigh waves, Bulk Acoustic Waves (BAW), and so on.

2.       The explanation of the results is questionable. It is unlikely to achieve such a low speed at the operating frequency as mentioned in the manuscript. All the eigenmodes on this surface demonstrate a significantly higher speed. What the authors have probably observed might just be an incidental occurrence due to electrical coupling. The referenced paper in their manuscript only shows a slow speed at a much higher frequency, around 10GHz, potentially caused by the same reason—electrical coupling. The difference between using gold (Au) and aluminum (Al) is negligible. The slowest speed achievable through material modification is by coating the entire surface with an ideal metal and shorting it. Still, the wave speed remains higher. The authors can compute this speed using the material’s properties including stress and strain tensors and density or via a COMSOL simulation. They will find the speed is still higher than what is presented in the manuscript.

Author Response

We thank the Reviewer for the comments. Below is a point-by-point reply to the Reviewer's comments. The modifications to the text are highlighted in yellow in the revised version of the manuscript.

  1. Unfortunately, we regret to inform the Reviewer that the comments provided are not sufficiently comprehensive for us to modify the manuscript. However, we did implement the Reviewer's previous suggestions and conducted additional simulations, which validated our understanding of the underlying physical phenomena of our study. It is worth noting that COMSOL incorporates built-in functions and variables specifically designed for calculating the S11 response. Consequently, performing such simulations aligns perfectly with the software's intended scope.

  2. Our explanation of the physical phenomena involved is rooted in the wave scattering theory, supported by the simulation results conducted based on the Reviewer's recommendations and supplemented by experimental data, all presented in the manuscript. The general agreement between the simulations and experiments does not reveal any evident systematic errors in our approach. Moreover, the Reviewer's claim that Al or Au fingers may lead to negligible differences in the SAW speed is erroneous. In fact, the density of Au is nearly ten times higher than that of Al, which causes a significant red-shift of the resonance frequency of the devices, as widely shown in the literature. Nevertheless, we believe that further investigations would be beneficial in gaining a deeper understanding of all the mechanisms at play, as pointed out in the conclusions.

Reviewer 3 Report

I consider that the manuscript could be accepted in its present form

Author Response

(The authors gave the same response as above.)

Reviewer 4 Report

First, the cited references relevant to the research in this manuscript are not enough. The research results of surface acoustic wave biosensors are extremely rich, and it is recommended that the author increase the citation of existing paper results.

Second, SAW devices can work in the ultra-highfrequency (UHF) range, commonly identified in the range 300 MHz – 3 GHz.This expression is not entirely accurate. In fact, for SAW devices, the general operating frequency is from tens of MHz to several GHz, which does not need to be defined as Super high frequency as radio frequency band. It is suggested that the author delete the expression and definition of Super high frequency.

Author Response

We thank the Reviewer for the comments. Below is a point-by-point reply to the Reviewer's comments. Modifications to the text are highlighted in yellow in the revised manuscript.

  1. The literature cited in our manuscript was carefully selected among the several SAW device works. We find all the cited papers relevant for introducing our study and supporting our methods and conclusions. As previously stated, we would gladly include other works that the Reviewer may suggest if pertinent to our study.
  2. Following the Reviewer's suggestion, we removed the UHF acronym from the revised manuscript's title and abstract.

Round 3

Reviewer 4 Report

The author answered well for the comments.